# Synergistic Cytotoxicity of Histone Deacetylase and Poly-ADP Ribose Polymerase Inhibitors and Decitabine in Breast and Ovarian Cancer Cells: Implications for Novel Therapeutic Combinations

**DOI:** 10.3390/ijms25179241

**Published:** 2024-08-26

**Authors:** Benigno C. Valdez, Apostolia M. Tsimberidou, Bin Yuan, Mehmet A. Baysal, Abhijit Chakraborty, Clark R. Andersen, Borje S. Andersson

**Affiliations:** 1Department of Stem Cell Transplantation and Cellular Therapy, The University of Texas MD Anderson Cancer Center, 1515 Holcombe Blvd., Houston, TX 77030, USA; 2Department of Investigational Cancer Therapeutics, The University of Texas MD Anderson Cancer Center, 1515 Holcombe Blvd., Houston, TX 77030, USAachakraborty3@mdanderson.org (A.C.); 3Department of Biostatistics, The University of Texas MD Anderson Cancer Center, Houston, TX 77030, USA

**Keywords:** breast cancer, ovarian cancer, HDAC inhibitors, PARP inhibitors, decitabine, DNA repair, synergistic cytotoxicity

## Abstract

Breast and ovarian cancers pose significant therapeutic challenges. We explored the synergistic cytotoxicity of histone deacetylase inhibitors (HDACis), poly(ADP-ribose) polymerase inhibitors (PARPis), and decitabine in breast (MDA-MB-231 and MCF-7) and ovarian (HEY-T30 and SKOV-3) cancer cell lines that were exposed to HDACi (panobinostat or vorinostat), PARPi (talazoparib or olaparib), decitabine, or their combinations. HDACi, PARPi, and decitabine combinations had synergistic cytotoxicity (assessed by MTT and clonogenic assays) in all cell lines (combination index < 1). Clonogenic assays confirmed the sensitivity of breast and ovarian cancer cell lines to the three-drug combinations (panobinostat, talazoparib, and decitabine; panobinostat, olaparib, and decitabine; vorinostat, talazoparib, and decitabine; vorinostat, olaparib, and decitabine). Cell proliferation was inhibited by 48–70%, and Annexin V positivity was 42–59% in all cell lines exposed to the three-drug combinations. Western blot analysis showed protein PARylation inhibition, caspase 3 and PARP1 cleavage, and c-MYC down-regulation. The three-drug combinations induced more DNA damage (increased phosphorylation of histone 2AX) than the individual drugs, impaired the DNA repair pathways, and altered the epigenetic regulation of gene expression. These results indicate that HDACi, PARPi, and decitabine combinations should be further explored in these tumor types. Further clinical validation is warranted to assess their safety and efficacy.

## 1. Introduction

Breast and ovarian cancers are major contributors to cancer-related mortality in women worldwide, highlighting the urgent need for novel therapeutic strategies to combat resistance and improve patient outcomes [1]. The American Cancer Society estimates that in the United States, 310,720 new cases of invasive breast cancer will be diagnosed in women in 2024, along with an additional 56,500 new cases of ductal carcinoma in situ (DCIS), and approximately 42,250 women will die from breast cancer [2]. Although breast cancer is more common in women, men can also rarely develop this disease, accounting for less than 1% of all breast cancer cases [3]. For ovarian cancer, 19,680 new diagnoses and 12,740 deaths are estimated in 2024 [4]. Epigenetic dysregulation, including aberrant histone modifications and DNA methylation, has been implicated in the pathogenesis and progression of both of these cancers [5]. Histone deacetylase inhibitors (HDACis) and poly(ADP ribose) polymerase inhibitors (PARPis) have emerged as promising agents for targeting epigenetic alterations and DNA repair pathways in cancer cells [6]. HDACis inhibit histone deacetylases, which modulate gene expression by regulating the accumulation of acetylated histones [7]. PARPis inhibit the DNA repair pathway in cells with deficiencies in homologous recombination, such as those with *BRCA* mutations, leading to synthetic lethality in cancer cells [8]. Decitabine inhibits DNA methyltransferase, which has shown potential in reversing epigenetic silencing and reactivating tumor suppressor genes in different types of cancer [9,10,11]. Recent studies have revealed that combining decitabine with other drugs that affect DNA and epigenetics can increase gene re-expression and drug sensitivity in cancer cells in vivo [12,13,14]. Moreover, decitabine has demonstrated clinical efficacy in treating myelodysplastic syndromes, with evidence of gene target expression modulation by demethylation and lower toxicity compared to traditional cancer chemotherapies [15].

The combination of HDACis, PARPis, and decitabine targets distinct but interconnected pathways involved in cancer progression and resistance [10,11,12]. By combining these three classes of drugs, their individual mechanisms of action can complement each other, leading to enhanced anti-cancer effects. For instance, HDACis can potentially make cancer cells more sensitive to PARPis by regulating the expression of genes involved in DNA repair and cell survival. Similarly, the re-expression of silenced genes by decitabine may work in synergy with the cytotoxic effects of PARPis, leading to a greater elimination of cancer cells [11].

The combinations of PARPis with HDACis are of particular interest in ovarian and breast cancers. Both preclinical and clinical studies have shown that HDACis and PARPis can synergize to increase efficacy in these malignancies [16,17]. In addition, the combination of HDACis and decitabine has been associated with synergistic cytotoxicity in ovarian cancer cells [18] and has demonstrated the potential to re-express silenced tumor suppressor genes and increase the effectiveness of conventional therapies in ovarian cancer [19].

In this study, we investigated the synergistic cytotoxicity of the combinations of HDACis, PARPis, and decitabine in breast and ovarian cancer cell lines and their potential mechanisms of antitumor activity.

## 2. Results

### 2.1. Sensitivity of Breast and Ovarian Cancer Cell Lines to HDACis, PARPis, and Decitabine

Dose–response experiments were performed using the *BRCA1* and *BRCA2* wild-type breast (MDA-MB-231 and MCF-7) and ovarian (HEY-T30 and SKOV-3) cancer cell lines to determine the differences in their drug sensitivity and the concentrations appropriate for the drug combination experiments. Figure 1 shows the proliferation of cells exposed to individual drugs and the calculated IC_50_ values. MDA-MB-231 and MCF-7 cells had similar sensitivity to vorinostat, talazoparib, and olaparib; MDA-MB-231 cells were more sensitive to panobinostat and decitabine compared to the MCF-7 cells (Figure 1A, Appendix A). The HEY-T30 cells were more sensitive to vorinostat, olaparib, and decitabine than the SKOV-3 cells, but the two cell lines had similar sensitivity to panobinostat and talazoparib (Figure 1B, Appendix A).

### 2.2. Synergistic Cytotoxicity of HDACis, PARPis, and Decitabine in Breast and Ovarian Cancer Cell Lines

To investigate whether HDACis, PARPis, and decitabine would exhibit synergistic cytotoxicity in breast and ovarian cancer cell lines, cells were exposed to various concentrations of single agents or combinations of the three drugs (HDACi plus PARPi plus decitabine), using a constant ratio, followed by the MTT assay. Figure 2 shows the calculated combination index at increasing drug effects using the three-drug combinations. Significant synergism was noted between HDACis, PARPis, and decitabine (combination index values < 1 at fractions affected > 0.5) in all cell lines.

Clonogenic assays were performed to further determine the synergistic cytotoxicity of HDACi, PARPi, and decitabine in breast and ovarian cancer cell lines. Figure 3A shows the colony formation in the plates, and Figure 3B shows the respective quantitative analysis results. Colony formation for the MDA-MB-231 breast cancer cell line was inhibited by 64% to 82% relative to the control when cells were exposed to the three-drug combinations (panobinostat, talazoparib, and decitabine; panobinostat, olaparib, and decitabine; vorinostat, talazoparib, and decitabine; vorinostat, olaparib, and decitabine). The MCF-7 cell line was less sensitive than the MDA-MB-231 cell line to these drug combinations as shown by 45% to 58% inhibition of colony formation (Figure 3B, Appendix A). However, none of these inhibitions were statistically significant.

In addition, colony formations for HEY-T30 and SKOV-3 ovarian cancer cell lines were inhibited by 50% to 98% and 64% to 92%, respectively, when cells were exposed to the three-drug combinations (Figure 3). The combinations of panobinostat, talazoparib, and decitabine and vorinostat, talazoparib, and decitabine showed some inhibition of the colony formation of HEY-T30 cells; a similar result was obtained when SKOV-3 cells were exposed to the combinations of panobinostat, talazoparib, and decitabine and panobinostat, olaparib, and decitabine (Figure 3B, Appendix A).

### 2.3. Drug-Mediated Inhibition of Cell Proliferation and PARylation, and Effects on Survival and Apoptosis Protein Markers

The results of the clonogenic assay (Figure 3) were consistent with those of the MTT assay for cell proliferation (Figure 4A). The addition of the hypomethylating agent decitabine to panobinostat plus talazoparib resulted in ~49%, ~50%, ~70%, and ~56% inhibition of cell proliferation in the MDA-MB-231, MCF-7, HEY-T30, and SKOV-3 cell lines, respectively; the addition of decitabine to panobinostat plus olaparib resulted in ~47%, ~49%, ~67%, and ~54% inhibition in the MDA-MB-231, MCF-7, HEY-T30, and SKOV-3 cell lines, respectively (Figure 4A). Similar results were obtained when decitabine was combined with vorinostat plus talazoparib, which caused ~48%, ~49%, ~69%, and ~52% inhibition of proliferation, and when decitabine was added to vorinostat plus olaparib, which resulted in ~48%, ~52%, ~70%, and ~50% inhibition of cell proliferation, in the MDA-MB-231, MCF-7, HEY-T30, and SKOV-3 cell lines, respectively.

Similar results were obtained when cells were exposed to the three-drug combinations and analyzed for programmed cell death using the Annexin V assay. The Annexin V positivity ranges in the MDA-MB-231, MCF-7, HEY-T30, and SKOV-3 cells were ~47–59%, ~44–49%, ~43–49%, and ~42–44%, respectively (Figure 4A, Appendix A). Table 1 shows the drug-mediated inhibition of cell proliferation (MTT assay) and activation of apoptosis (Annexin V assay), comparing the effects of each drug combination with the individual drugs in the breast cancer (MDA-MB-231 and MCF-7) and ovarian cancer (HEY-T30 and SKOV-3) cell lines.

### 2.4. The Three-Drug Combinations Inhibit PARylation and Enhance Cleavage of Caspase 3 and PARP1

Next, we investigated the effects of PARPis, HDACis, and decitabine on PARylation. As talazoparib and olaparib are potent PARPis, we investigated whether they also inhibit the protein PARylation. The HDACis panobinostat and vorinostat did not inhibit PARylation, whereas the PARPis talazoparib and olaparib decreased the PARylation levels in all cell lines, and the addition of decitabine enhanced their inhibitory effects (Figure 4B). Surprisingly, decitabine alone also decreased the levels of PARylation in the four cell lines. As expected, the HDACis panobinostat and vorinostat increased the levels of acetylated proteins including histone 3 and α-tubulin when used individually, and the effects were further enhanced when they were combined with a PARPi and decitabine (Figure 4B). To further investigate whether the decrease in cell proliferation and increase in Annexin V positivity (Figure 4A) were associated with apoptosis, we analyzed the cleavage of caspase 3 and PARP1. The triple-drug combinations markedly enhanced both caspase 3 and PARP1 cleavages in all cell lines (Figure 4B). Cells exposed to the triple-drug combinations exhibited increased phosphorylation of histone 2AX (γ-H2AX). This finding indicates DNA damage response (double-strand break formation and/or activation) likely contributed to the activation of nuclear DNases, mediated by caspases. These observations are consistent with a decreased level of pro-survival c-MYC in cells exposed to HDACi plus PARPi plus decitabine (Figure 4B).

### 2.5. HDACi, PARPi, and Decitabine Combinations Affect the Levels of Proteins Involved in DNA Damage Response and Repair

Post-translational modifications (acetylation and PARylation) of proteins associated with DNA repair may affect their stability, as previously described for BRCA1 and UHFR1 [20,21]. When all cell lines were exposed to the three-drug combinations, there was a decrease in the levels of the ATM (ataxia–telangiectasia mutated) protein, which is responsible for DNA double-strand break repair and cell cycle checkpoint activation. Additionally, there was a decrease in the level of BRCA1, which is involved in homologous recombination at the level of ATRX, a chromatin remodeling protein that participates in homologous recombination repair (Figure 5). In cells exposed to the three-drug combinations, a decrease in the levels of the non-homologous end-joining repair proteins DNAPKcs, Artemis, and DNA ligase 1 was also noted. Moreover, the phosphorylation of DNAPKcs at serine 2056 increased in cells exposed to three-drug combinations. Analysis of some of the subunits of the nucleosome remodeling and deacetylase (NuRD) complex showed decreased levels of the CHD3, CHD4, MBD3, MTA1, and HDAC1 proteins in all cell lines exposed to the three-drug combinations (Figure 5). Overall, these results demonstrated that the three-drug combinations decreased the levels of proteins involved in DNA damage response.

## 3. Discussion

This preclinical study provides evidence of the synergistic interactions between HDACi (panobinostat or vorinostat) and PARPi (talazoparib or olaparib) and the hypomethylating agent decitabine in breast (MDA-MB-231 and MCF-7) and ovarian (HEY-T30 and SKOV-3) cancer cell lines and demonstrates the clinical potential of using a drug regimen that targets epigenetic modifications and DNA repair inhibition in breast and ovarian tumors (Figure 2 and Figure 5). HDACis and decitabine are epigenetic regulators known to increase DNA damage and downregulate key DNA repair proteins [22,23,24,25], which make cancer cells highly dependent on PARP for DNA repair and therefore very sensitive to combined HDACis, PARPis, and decitabine.

A surprising finding in the present study was the HDACi-mediated inhibition of PARylation only in MCF-7 cells and not in the remaining cell lines (Figure 4B). This is in contrast with our previous observation that the HDACis romidepsin and vorinostat, used individually, inhibited protein PARylation in acute myeloid leukemia, T-cell acute lymphoblastic leukemia, diffuse large B-cell lymphoma, and multiple myeloma cell lines and in mononuclear cells from patients with various types of leukemia [25]. This lack of agreement between the studies may be attributed to differences in the sensitivity of the cell lines to drug concentrations that were determined taking into consideration the IC_50_ values. However, the combinations of HDACi, PARPi, and decitabine inhibited protein PARylation across the four cell lines (Figure 4B).

The three-drug combinations also affected the levels of proteins involved in DNA damage response and repair, leading to decreased levels of proteins associated with DNA repair and chromatin remodeling (Figure 5). These results may relate to the drug combinations’ inhibition of protein PARylation. It is known that PARylation modulates the stability and activity of proteins involved in DNA damage response [26,27]. With the observed drug-mediated DNA damage, as indicated by the increased level of γ-H2AX in cells exposed to the three-drug combinations (Figure 4B), DNA repair was expected to be compromised, which committed cells to undergo apoptosis.

Notably, the components of the NuRD complex were down-regulated in cells exposed to the three-drug combinations (Figure 5). NuRD is known to play a crucial role in chromatin remodeling and deacetylation processes [28] and controls DNA damage signaling and repair [29]. The observed decrease in the NuRD subunits—the novel regulators of DNA damage response—may have resulted in increased DNA damage and/or increased expression of tumor suppressor genes including *p21*, as has been reported for human osteosarcoma and *BRCA*-proficient breast cancer cell lines [29,30].

The three-drug combinations also decreased the levels of proteins involved in non-homologous DNA end-joining (Figure 5). However, the phosphorylation of DNAPKcs at serine 2056 notably increased (Figure 5) and may have resulted in the inactivation of its kinase activity and DNA repair functions as previously reported [31,32].

Our results are consistent with studies demonstrating the synergistic cytotoxicity of HDACis and PARPis [33,34]. It has been reported that increased acetylation of histone 3 at lysine 9 blocked its ADP-ribosylation at serine 10, suggesting an antagonistic relationship between acetylation and PARylation [33]. Others have reported that the HDACi trichostatin A increased protein acetylation and trapped PARP1 to double-strand DNA breaks in the K562 leukemia cell line, and the combination of HDACi with talazoparib resulted in a dose-dependent increase in PARP1 trapping, which correlated with apoptosis [34].

It has been reported that in breast cancer the efficacy of combining PARPi with HDACi depends on the *BRCA* mutation status and the tumor microenvironment [35]. Additionally, homologous recombination deficiency scores have been found to correlate strongly with *BRCA1/2* deficiency, regardless of breast cancer subtype [36]. In vitro studies using benzamide derivatives of olaparib, which have both PARP and HDAC inhibition activities, showed induction of BRCAness, promotion of cytosolic DNA accumulation, activation of the cGAS–STING pathway, induction of DNA damage, and production of proinflammatory chemokines through the JAK–STAT pathway in triple-negative breast cancer cell lines [37].

With the downregulation of BRCA1, it is conceivable that the expression and activity of low-fidelity polymerases might be upregulated to compensate for the impaired DNA repair and could contribute to genomic instability and resistance [38]. Preclinical evidence has provided a rationale for targeting DNA damage response pathways by combining PARPi with HDACi as a mechanism for reducing homologous recombination efficiency in ovarian cancer [39]. The combination of the DNA hypomethylating agent guadecitabine and the PARPi talazoparib has been effective in inhibiting breast and ovarian cancers regardless of *BRCA* mutations, indicating the need for further clinical exploration of this drug combination in PARPi-resistant cancers [17].

Recent studies have highlighted the potential of PARP inhibitors beyond DNA repair, providing a scientific rationale for combining PARPi with anti-PD-L1 therapy in DNA repair-deficient populations [40]. In ovarian cancer, the combination of PARPi with HDACi has been found to improve the efficacy of PARPi-based immunotherapy by enhancing the homologous recombination deficiency functional phenotype, thus overcoming resistance to immunotherapy [16]. These studies indicate that PARP inhibitors have broader implications in cancer therapy, extending beyond their role in DNA repair mechanisms. HDACis are known to increase IL-8 expression in ovarian cancer cells [41], resulting in their increased survival and proliferation. Further research should address whether PARPis and/or decitabine can reverse this process.

The current study has limitations, which should be considered prior to conducting clinical trials. A three-dimensional cell model may be more effective in predicting anti-tumor efficacy. Drug toxicities and pharmacodynamics cannot be determined using cell line models, and effects on normal cells should be considered. Cell cycle analyses to determine how the drug combinations affect cell cycle progression and contribute to the cytotoxic effects [42] were not performed. Theoretically, the drug synergy in normal immortalized cell lines such as MCF10 (a non-tumorigenic epithelial cell line) and normal immortalized ovarian cell lines should be tested to determine if the observed synergistic effects are specific to cancer cells. This could help confirm the selectivity of the drug combinations for cancer cells over normal cells. However, it is also conceivable that the results obtained in immortalized cancer cell lines should be corroborated by showing that immortalized normal tissue lines are less sensitive to the drug combinations. Subsequently, this raises questions about differences in drug metabolism between normal and malignant cells in a clinical scenario, such that a beneficial therapeutic index can be achieved even if normal cells appear to be quite sensitive to the used combinations in vitro. In vivo experiments with xenograft models from each cancer cell line were not performed. Although animal models are frequently used, they have their own limitations. They are costly and lengthy, and results derived from these studies cannot be directly extrapolated to humans, owing to differences in the metabolism of the study drugs between humans and animal models. Carefully designed phase I-II studies are still required to confirm if the obtained results are clinically meaningful. The cytotoxicity experiments focused on the three-drug combinations. The rationale for focusing on the three-drug combinations is based on the concept that these combinations vs. the double-drug combinations might optimize antitumor effects, minimizing adverse events associated with the study drugs in a phase I clinical trial (that would require low doses of each drug to prevent toxicities).

While our study provides evidence of c-MYC downregulation and DNA repair impairment, the specific roles of AXL and low-fidelity polymerases in this context warrant further investigation. The critical role of AXL, a receptor tyrosine kinase, in promoting tumor cell survival, metastasis, and therapeutic resistance is well established [43,44]. AXL activation can lead to the generation of persister cells, a subpopulation of cancer cells that can survive treatments and potentially cause tumor recurrence [43,45,46,47]. Therefore, AXL may influence the response to combination therapy involving HDACis, PARPis, and decitabine. AXL signaling pathways could potentially interact with the mechanisms targeted by these drugs, affecting their efficacy in inhibiting cell proliferation, inducing apoptosis, and impairing DNA repair pathways. Similarly, the presence of low-fidelity polymerases could further disrupt DNA repair mechanisms targeted by PARPis, potentially enhancing the cytotoxic effects of the combination treatment. Furthermore, low-fidelity polymerases such as Pol η, Pol ι, Pol κ, and POLQ, which are involved in translesion DNA synthesis, may introduce mutations owing to their error-prone nature, further compromising DNA repair mechanisms and contributing to genomic instability [48,49,50]. In the presence of c-MYC downregulation, the reliance on low-fidelity polymerases may increase, leading to a higher tumor mutational burden, cancer progression, and resistance to treatment. Further research is needed to elucidate the role of AXL signaling, low-fidelity polymerase activity, and impaired DNA repair for developing targeted therapies in the clinical setting.

In conclusion, the current in vitro study provides proof of concept and focuses on the mechanisms of the synergistic cytotoxicity of HDACis, PARPis, and decitabine in breast and ovarian cancer cell lines. The identified molecular mechanisms encompass the inhibition of cellular proliferation, the induction of apoptosis, and notable alterations in crucial proteins associated with DNA damage response and repair pathways. The study contributes to the growing body of evidence supporting the use of combination therapies targeting epigenetic modifications and DNA repair pathways as promising strategies for cancer treatment. The findings from this study provide a foundation for considering this three-drug combination for clinical development in breast and ovarian cancers. The observations from the current experiments provide the rationale to design phase I clinical trials to evaluate the safety and efficacy of these drug combinations in these tumor types, hoping to overcome the resistance mechanisms and improve the clinical outcomes of patients with these tumor types.

## 4. Materials and Methods

### 4.1. Cell Lines and Drugs

The breast cancer cell lines MDA-MB-231 (Catalog number HTB-26) and MCF-7 (HTB-22) and the ovarian cancer cell lines HEY-T30 (CRL-3252) and SKOV-3 (HTB-77) were purchased from American Type Culture Collection (ATCC; Manassas, VA, USA). All cells possess wild-type *BRCA1* and *BRCA2* genes. MDA-MB-231 is a triple-negative breast cancer cell line that lacks estrogen receptor, progesterone receptor, HER2, and E-cadherin but expresses mutated *p53* [51]. MCF-7 cells express estrogen receptor, progesterone receptor, HER2, and E-cadherin [52]. HEY-T30 and SKOV-3 are taxol-resistant and tumor necrosis factor (TNF)-resistant cell lines, respectively. Following ATCC protocols, all cells were cultured in a 5% CO_2_ humidified incubator at 37 °C. MDA-MB-231, MCF-7, and SKOV-3 cells were cultured in Dulbecco’s Modified Eagle Medium, while HEY-T30 cells were grown in Roswell Park Memorial Institute 1640 medium. Both media contained 10% heat-inactivated fetal bovine serum along with antibiotics (100 IU/mL penicillin and 100 μg/mL streptomycin).

The HDACis panobinostat and vorinostat, the PARPis talazoparib and olaparib, and the demethylating agent decitabine were obtained from Selleck Chemicals (Houston, TX, USA). Stock solutions were prepared using dimethyl sulfoxide, of which the final concentration did not exceed 0.1% of the total volume.

### 4.2. Determination of IC_50_ and Drug Synergism

Cell proliferation was determined using the 3-(4,5-dimethylthiazol-2-yl)-2,5-diphenyl tetrazolium bromide (MTT) assay. Briefly, 100 µL of cells (MDA-MB-231: 2.5 × 10^4^ cells/mL; MCF-7: 2.5 × 10^4^ cells/mL; HEY-T30: 0.9 × 10^4^ cells/mL; and SKOV-3: 2.5 × 10^4^ cells/mL) were seeded per well in a 96-well plate. After 24 h, the medium was replaced with 100 µL of appropriate medium containing drug(s), and cells were incubated for 3 days, after which the medium was replaced with fresh medium without drugs; this was aimed at mimicking a clinical scenario, where patients usually are exposed to drugs for a defined period of time, followed by a rest period. The MTT assay was performed by adding 30 µL of MTT (2 mg MTT/mL) in phosphate-buffered saline (PBS) per well and incubating the plates for 4 h at 37 °C. The insoluble purple formazan product was dissolved by adding 100 μL of solubilization solution (0.1 N HCl in isopropanol containing 10% Triton X-100) to each well, mixing, and incubating the plates at 37 °C overnight. Absorbance at 570 nm was measured using a Victor X3 plate reader (Perkin Elmer Life and Analytical Sciences, Shelton, CT, USA). The rate of cell proliferation was determined relative to the control cells exposed to solvent alone. The IC_50_ values were calculated using CalcuSyn 2.0 software (ComboSyn, Inc., Paramus, NJ, USA) as previously described [53].

To determine drug synergism, cells were seeded in 96-well plates as described above. The medium was changed after 24 h, and the cells were exposed to various drug combinations at a constant ratio of concentrations for 3 days prior to the MTT assay. Fractions affected refer to cell death, which was determined using the MTT assay. Drug combination effects were estimated based on the combination index values calculated using CalcuSyn 2.0 software.

### 4.3. Colony Formation/Clonogenic Assay

MDA-MB-231 (1.4 × 10^3^ cells), MCF-7 (1.6 × 10^3^ cells), HEY-T30 (0.4 × 10^3^ cells), and SKOV-3 (0.7 × 10^3^ cells) cells were seeded (3 mL) onto six-well plates. The next day, the medium was replaced with fresh medium containing drug(s), and the cells were incubated at 37 °C for 3 days. Then, the medium was replaced with fresh medium without drugs. After 1–2 weeks, formed colonies were fixed using 4% glutaraldehyde for 20 min. The colonies were then washed thrice with PBS and stained using 0.5% crystal violet for 15 min. Excess stain was removed by washing twice with PBS. The procedures were performed at room temperature, and the experiments were repeated at least three times.

### 4.4. MTT Assay and Western Blot Analysis

Cells (MDA-MB-231: 4.2 × 10^4^ cells/mL; MCF: 7: 5 × 10^4^ cells/mL; HEY-T30: 2.5 × 10^4^ cells/mL; SKOV: 3: 5 × 10^4^ cells/mL) were seeded (6 mL) in T25 flasks overnight. The next day, the old medium was replaced with fresh medium containing drug(s), and the cells were exposed continuously to drug(s) for 3 days. Cells were dissociated from the flask using accutase (MilliporeSigma, St. Louis, MO, USA), harvested, and washed with cold PBS. The cell proliferation rate was measured using the MTT assay, as described above. Programmed cell death was determined by flow cytometric measurements of phosphatidylserine externalization with Annexin-V-FLUOS (Roche Diagnostics, Indianapolis, IN, USA) and 7-aminoactinomycin D (BD Biosciences, San Jose, CA, USA) using a Muse Cell Analyzer (MilliporeSigma, St. Louis, MO, USA). Briefly, 50 µL of cell suspension was combined with 50 µL of Annexin V reagent, incubated at room temperature for 20 min, and analyzed using a Muse Cell Analyzer.

For Western blot analysis, cells were lysed with lysis buffer (Cell Signaling Technology, Danvers, MA, USA). Western blot analysis was performed as previously described [25]. The β-actin protein was used as an internal control. Antibodies used for immunoblotting are described in the Appendix A.

### 4.5. Drug Concentrations for Colony Formation/Clonogenic Assay, MTT Assay, and Western Blot Analysis

For the double- and triple-drug combinations, the drug concentrations were the same as indicated in the single-drug concentrations for the same experiment.

### 4.6. Statistical Analysis

Analyses were performed independently for breast and ovarian cell lines. Mixed-effect analysis of variance was used to model cellular proliferation percentage (where the percentage was referenced to the zero-dose sample) and dose (with 6 discrete doses, excluding 0), separately by cell line, clustering on study day. Contrasts were used to assess model-adjusted differences between dose levels using the “emmeans” package [54], with Tukey-adjusted *p*-values. For cytotoxicity, the association between proliferation and drug treatment was modeled similarly, separately by assay (MTT versus Annexin V), and differences were assessed by contrasts with Hommel-adjusted *p*-values for comparisons between three-drug combinations and their component drugs. Analysis of variance was used for colony formation modeling relating proliferation to drug treatment, with differences assessed by contrasts with Hommel-adjusted *p*-values for comparisons between two-drug and three-drug combinations and their component drugs. Mixed-effect modeling was performed using the “nlme” package [55,56]. All statistical modeling of proliferation was conducted using R statistical software (version 4.3.1), with an assumption of 95% level of statistical confidence.

## Figures and Tables

**Figure 1 ijms-25-09241-f001:**
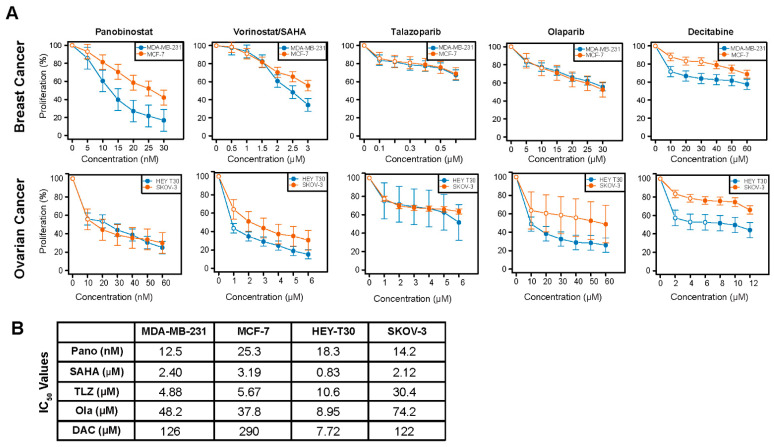
Dose–response curves for the studied drugs in breast and ovarian cancer cell lines. Cells were seeded in 96-well plates overnight and exposed to different concentrations of individual drugs for 3 days as described in Methods. Rate of cell proliferation was determined relative to control by MTT assay (**A**,**B**). Model-adjusted means are shown with 95% confidence intervals for the non-zero doses modeled, and solid points indicate a significant difference from the first non-zero dose (Appendix A). Each cell line of each drug was modeled independently. IC_50_ values were determined using CalcuSyn 2.0 software. Pano: panobinostat; SAHA: vorinostat; TLZ: talazoparib; Ola: olaparib; DAC: decitabine.

**Figure 2 ijms-25-09241-f002:**
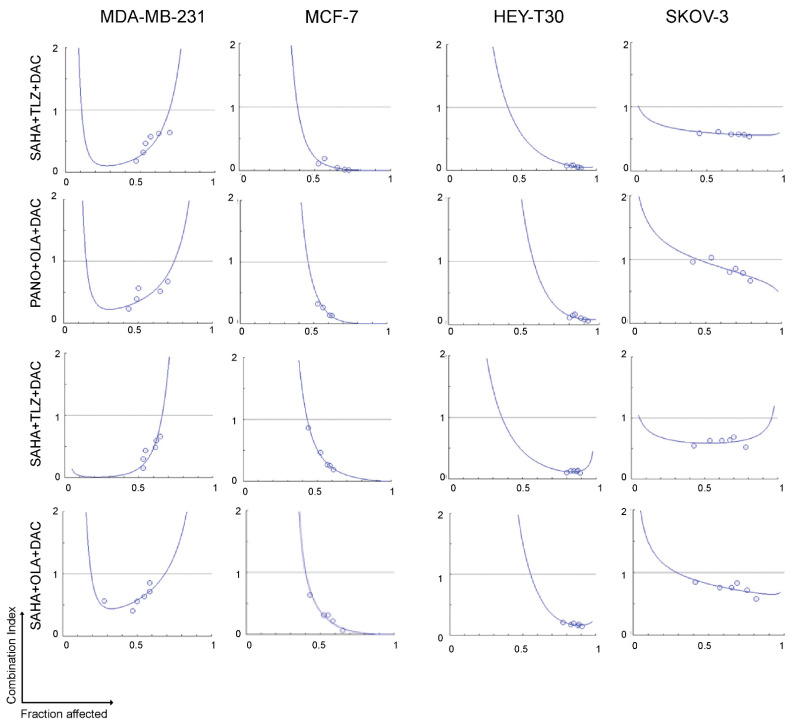
Synergistic cytotoxicity of HDACi, PARPi, and decitabine. Cells were seeded in 96-well plates overnight and exposed to drugs individually, or in three-drug combinations at a constant concentration ratio, and cell proliferation was analyzed after 3 days. The relationships between the calculated combination indexes (*Y*-axis) and fractions affected (*X*-axis) are shown. A combination index < 1.0 indicates synergism. The graphs are representative of two independent experiments.. SAHA: vorinostat; TLZ: talazoparib; DAC: decitabine; PANO: panobinostat; OLA: olaparib.

**Figure 3 ijms-25-09241-f003:**
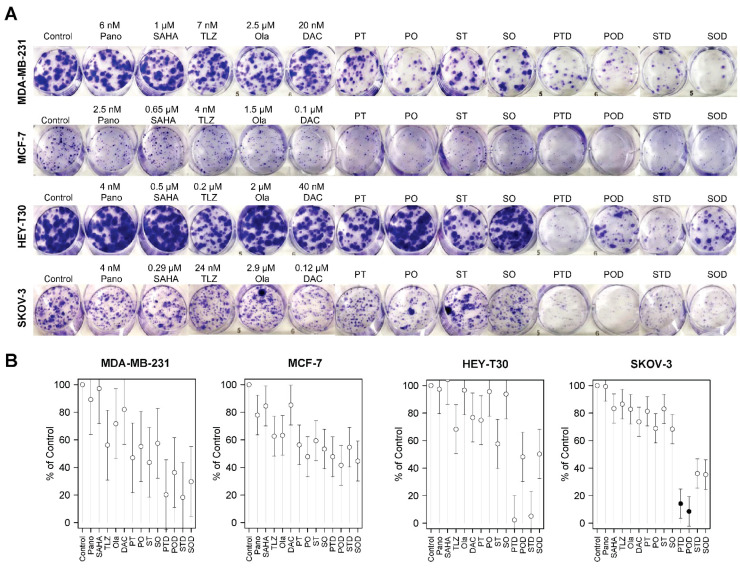
Colony formation assay. Cells were seeded in six-well plates overnight and exposed to individual drugs or three-drug combinations for 1–2 weeks and stained as described in Methods (**A**). Colony formation is presented relative to control (**B**). For the double- and triple-drug combinations, the drug concentrations were the same as indicated in the single-drug concentrations. Model-adjusted means are shown with 95% confidence intervals, and solid points indicate a significant synergistic difference from all of the component drugs (see Appendix A). Each cell line of each drug was modeled independently. Number of replicates = 3; Pano/P: panobinostat; SAHA/S: vorinostat; TLZ/T: talazoparib; Ola/O: olaparib; DAC/D: decitabine.

**Figure 4 ijms-25-09241-f004:**
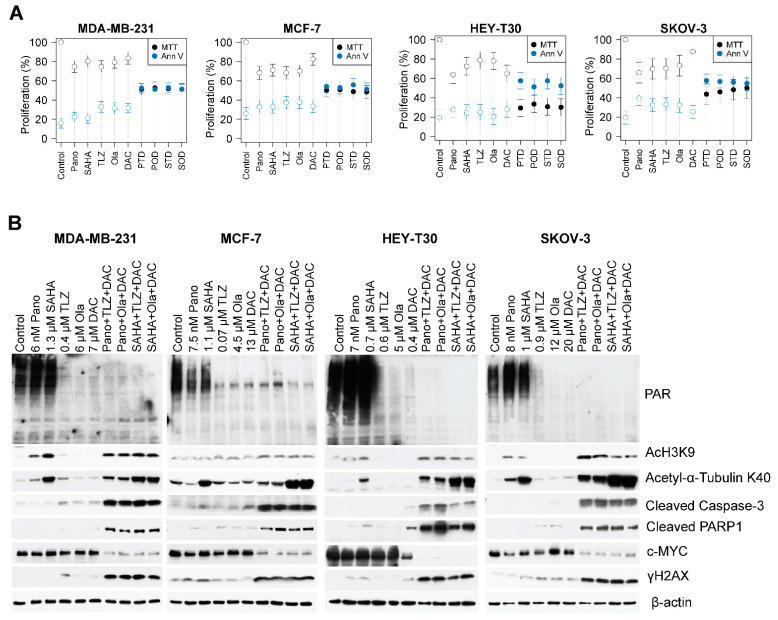
Drug-mediated inhibition of cell proliferation and PARylation and effects on survival and apoptosis protein markers. Cells were seeded in T25 flasks overnight and exposed to individual drugs or three-drug combinations for 3 days, harvested, and analyzed for cell proliferation by MTT assay (**A**) and Western blotting (**B**). For the double- and triple-drug combinations, the drug concentrations were the same as indicated in the single-drug concentrations. Model-adjusted means are shown with 95% confidence intervals, and solid points indicate a significant synergistic difference from all of the component drugs (Appendix A). Each cell line of each drug was modeled independently. Number of replicates = 3; Pano/P: panobinostat; SAHA/S: vorinostat; TLZ/T: talazoparib; Ola/O: olaparib; DAC/D: decitabine.

**Figure 5 ijms-25-09241-f005:**
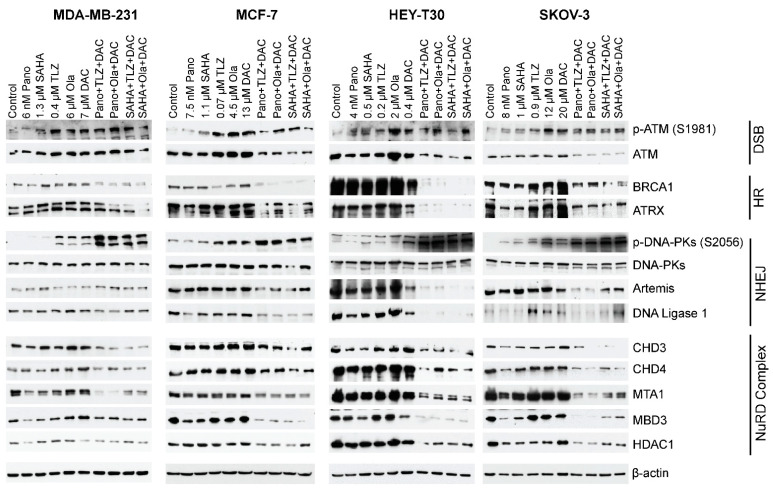
Effects of drugs on the levels of various proteins involved in DNA repair/DNA damage response. Cells were exposed to the indicated drug concentrations for 3 days prior to analysis by Western blotting. For the double- and triple-drug combinations, the drug concentrations were the same as indicated in the single-drug concentrations. Number of replicates = 3; DSB: double-strand break; HR: homologous recombination; NHEJ: non-homologous DNA end-joining; NuRD: nucleosome remodeling and deacetylase. Pano: panobinostat; SAHA: vorinostat; TLZ: talazoparib; Ola: olaparib; DAC: decitabine.

**Table 1 ijms-25-09241-t001:** Comparison of the effects of each drug combination with the individual drugs in the breast (MDA-MB-231 and MCF-7) and ovarian (HEY-T30 and SKOV-3) cancer cell lines in the drug-mediated inhibition of cell proliferation (MTT assay) and activation of apoptosis (Annexin V assay). *p* ≤ 0.05 is considered statistically significant.

**Breast Cancer Cell Lines**										
**MDA-MB-231**						**MCF-7**				
	**Pano**	**SAHA**	**TLZ**	**OLA**	**DAC**	**Pano**	**SAHA**	**TLZ**	**OLA**	**DAC**
**Cell Proliferation**										
**PTD**	<0.0001		<0.0001		<0.0001	0.003		0.003		<0.0001
**POD**	<0.0001			<0.0001	<0.0001	0.003			0.0009	<0.0001
**STD**		<0.0001	<0.0001		<0.0001		0.0001	0.001		<0.0001
**SOD**		<0.0001		<0.0001	<0.0001		<0.0001		0.0002	<0.0001
**Apoptosis**										
**PTD**	<0.0001		0.0001		<0.0001	0.011		0.068		0.014
**POD**	<0.0001			0.0001	0.0001	0.021			0.1	0.022
**STD**		<0.0001	0.0001		<0.0001		0.004	0.031		0.005
**SOD**		<0.0001		0.0001	0.0001		0.036		0.13	0.046
**Ovarian Cancer Cell Lines**										
**HEY-T30**						**SKOV-3**				
	**Pano**	**SAHA**	**TLZ**	**OLA**	**DAC**	**Pano**	**SAHA**	**TLZ**	**OLA**	**DAC**
**Cell Proliferation**										
**PTD**	<0.0001		<0.0001		<0.0001	0.0004		<0.0001		<0.0001
**POD**	<0.0001			<0.0001	<0.0001	0.0008			<0.0001	<0.0001
**STD**		<0.0001	<0.0001		<0.0001		0.0005	0.0005		<0.0001
**SOD**		<0.0001		<0.0001	<0.0001		0.0008		0.0003	<0.0001
**Apoptosis**										
**PTD**	0.0008		0.0003		0.001	0.037		0.001		<0.0001
**POD**	0.0101			0.0006	0.01	0.037			0.002	<0.0001
**STD**		0.0002	0.0003		0.001		0.002	0.003		<0.0001
**SOD**		0.004		0.0006	0.01		0.006		0.008	0.0001

**Abbreviations:** DAC: decitabine; OLA: olaparib; Pano: panobinostat; POD: panobinostat plus olaparib plus decitabine; PTD: panobinostat plus talazoparib plus decitabine; SAHA: vorinostat; SOD: SAHA (vorinostat) plus olaparib plus decitabine; STD: SAHA (vorinostat) plus talazoparib plus decitabine; TLZ: talazoparib.

## Data Availability

Data is contained within the article and Appendix A.

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
