# Peer review of "Synergistic Cytotoxicity of Histone Deacetylase and Poly-ADP Ribose Polymerase Inhibitors and Decitabine in Breast and Ovarian Cancer Cells: Implications for Novel Therapeutic Combinations"

_ijms, 2024, doi:10.3390/ijms25179241_

Round 1
Reviewer 1 Report
Comments and Suggestions for Authors
This is a preclinical study that indicates a synergistic interaction between HDAC inhibitors (panobinostat or vorinostat), PARP inhibitors (talazoparib or olaparib), and the hypo-methylating agent decitabine in breast and ovarian cancer cell lines. Specifically, the authors analyze the effect of the above drugs on cell proliferation, cytotoxicity, colony formation ability, apoptosis and protein levels of several DNA damage response and repair genes in breast and ovarian cancer cells.
Though rather descriptive, the study provides a useful new information regarding the synergistic activity of the above drugs, suggesting their potential combination use in future clinical studies.
The specific points are summarized below.
1. Since HDAC inhibitors are known to increase IL-8 expression in ovarian cancer cells, resulting in their increased survival and proliferation, it would be important to test whether PARP inhibitors and/or decitabine can reverse this process.
Minor points:
2. The formatting of the Abstract (font) should be corrected.
3. The References numbers are duplicated.
Author Response
We would like to thank the reviewers for their insightful comments and constructive feedback.
Reviewer 1 Comments:
Comments and Suggestions for Authors
This is a preclinical study that indicates a synergistic interaction between HDAC inhibitors (panobinostat or vorinostat), PARP inhibitors (talazoparib or olaparib), and the hypo-methylating agent decitabine in breast and ovarian cancer cell lines. Specifically, the authors analyze the effect of the above drugs on cell proliferation, cytotoxicity, colony formation ability, apoptosis and protein levels of several DNA damage response and repair genes in breast and ovarian cancer cells.
Though rather descriptive, the study provides a useful new information regarding the synergistic activity of the above drugs, suggesting their potential combination use in future clinical studies.
The specific points are summarized below.
- Since HDAC inhibitors are known to increase IL-8 expression in ovarian cancer cells, resulting in their increased survival and proliferation, it would be important to test whether PARP inhibitors and/or decitabine can reverse this process.
Reply: We agree with the reviewer that it would be important to test whether PARP inhibitors and/or decitabine can reverse this process. We have revised the Discussion section to include this statement and added the relevant reference: “HDACis are known to increase IL-8 expression in ovarian cancer cells [41], resulting in their increased survival and proliferation. Further research should address whether PARPis and/or decitabine can reverse this process.“ (Page 11; line no. 300-302).
Unfortunately, this was not an endpoint of the study, and we do not have the resources to test this hypothesis at this time.
Minor points:
- The formatting of the Abstract (font) should be corrected.
Reply: Thank you for bringing this to our attention. The issue has been addressed.
- The References numbers are duplicated.
Reply: We have carefully reviewed the reference list and could not identify duplicated references. The references numbered 2 and 3, which may appear similar at first glance, refer to distinct web pages on the American Cancer Society website. Reference 2 refers to breast cancer in general, while reference 3 refers to breast cancer in men. If this reply does not address your comment, we would appreciate it if you could please specify the reference numbers you refer to.
- American Cancer Society. Key Statistics for Breast Cancer. Available online: https://www.cancer.org/cancer/types/breast-cancer/about/how-common-is-breast-cancer.html (accessed on 1/29/2024).
- American Cancer Society. Key Statistics for Breast Cancer in Men. Available online: https://www.cancer.org/cancer/types/breast-cancer-in-men/about/key-statistics.html (accessed on 1/29/2024).
Reviewer 2 Report
Comments and Suggestions for Authors
This study explores the synergistic cytotoxicity of combining histone deacetylase inhibitors (HDACis), poly(ADP-ribose) polymerase inhibitors (PARPis), and decitabine in breast (MDA-MB-231 and MCF-7) and ovarian (HEY-T30 and SKOV-3) cancer cell lines. Combinations of HDACis (panobinostat or vorinostat), PARPis (talazoparib or olaparib), and decitabine demonstrated significant synergistic effects, evidenced by reduced cell proliferation (48%-70%) and increased apoptosis (42%-59%). These combinations inhibited protein PARylation, promoted caspase 3 and PARP1 cleavage, down-regulated c-MYC, and induced more DNA damage than individual drugs. The 3-drug regimens impaired DNA repair pathways and altered epigenetic regulation of gene expression. These findings support further investigation of HDACi, PARPi, and decitabine combinations as potential treatments for breast and ovarian cancers, emphasizing the need for clinical validation to assess their safety and efficacy. The study is interesting. It is indeed important for patient care. however, the major missing is the animal studies.
Major Comments:
1. What is the status of synergy in MCF10 lines or normal ovarian immortalized lines? Is this synergy specific to cancer cells? The authors need to test the drug synergy in normal immortalized cell lines such as MCF10 (a non-tumorigenic epithelial cell line) and normal immortalized ovarian cell lines to determine if the observed synergistic effects are specific to cancer cells. This will help confirm the selectivity of the drug combinations for cancer cells over normal cells.
2. As BRCA1 is downregulated, have the authors checked for the involvement of low fidelity polymerases? Given the downregulation of BRCA1, the authors should investigate the expression and activity of low fidelity polymerases, which might be upregulated to compensate for the impaired DNA repair and could contribute to genomic instability and resistance.
3. Based on the evidence of c-MYC downregulation and DNA repair impairment, what is the role of AXL and low fidelity polymerases? The authors should discuss the potential roles of AXL and low fidelity polymerases in the context of their findings. AXL is known to be involved in cancer progression and resistance and generation of persister cells, while low fidelity polymerases may further compromise DNA repair mechanisms.
4. Authors have to validate the synergy effects in animal models? To enhance the scientific soundness of the study, the authors should conduct in vivo experiments using at least one xenograft model from each cancer cell line. Testing the drug combination that showed the strongest synergy in these models would provide valuable insights into the therapeutic potential of the combination treatments.
5. Authors have to analyze the impact of the drug synergy on the cell cycle?Given the strong synergistic effects observed, the authors should perform a cell cycle analysis to determine how the drug combinations affect cell cycle progression and contribute to the cytotoxic effects.
6. Authors have to include pvalues in all the graphs (Including in the figure will be helpful for the readers to understand the comparisons. Also include the number of replicates in the figure legend.
Minor Comments:
1. Ensure the correct spelling for c-MYC (not c-MYK) in figure 2B.
Author Response
Reviewer 2 Comments:
Comments and Suggestions for Authors
This study explores the synergistic cytotoxicity of combining histone deacetylase inhibitors (HDACis), poly(ADP-ribose) polymerase inhibitors (PARPis), and decitabine in breast (MDA-MB-231 and MCF-7) and ovarian (HEY-T30 and SKOV-3) cancer cell lines. Combinations of HDACis (panobinostat or vorinostat), PARPis (talazoparib or olaparib), and decitabine demonstrated significant synergistic effects, evidenced by reduced cell proliferation (48%-70%) and increased apoptosis (42%-59%). These combinations inhibited protein PARylation, promoted caspase 3 and PARP1 cleavage, down-regulated c-MYC, and induced more DNA damage than individual drugs. The 3-drug regimens impaired DNA repair pathways and altered epigenetic regulation of gene expression. These findings support further investigation of HDACi, PARPi, and decitabine combinations as potential treatments for breast and ovarian cancers, emphasizing the need for clinical validation to assess their safety and efficacy. The study is interesting. It is indeed important for patient care. however, the major missing is the animal studies.
Major Comments:
- What is the status of synergy in MCF10 lines or normal ovarian immortalized lines? Is this synergy specific to cancer cells? The authors need to test the drug synergy in normal immortalized cell lines such as MCF10 (a non-tumorigenic epithelial cell line) and normal immortalized ovarian cell lines to determine if the observed synergistic effects are specific to cancer cells. This will help confirm the selectivity of the drug combinations for cancer cells over normal cells.
Reply: To address this comment, we revised the Discussion section as follows: “Theoretically, the drug synergy in normal immortalized cell lines such as MCF10 (a non-tumorigenic epithelial cell line) and normal immortalized ovarian cell lines should be tested to determine if the observed synergistic effects are specific to cancer cells. This could help confirm the selectivity of the drug combinations for cancer cells over normal cells. However, it is also conceivable that the results obtained in immortalized cancer cell lines should be corroborated by showing that immortalized normal tissue lines are less sensitive to the drug combinations. Subsequently, this raises questions about differences in drug metabolism between normal and malignant cells in a clinical scenario, such that a beneficial therapeutic index can be achieved even if normal cells appear to be quite sensitive to the used combinations in vitro. (Page 11; line no. 309 – 318).
Unfortunately, although we recognize the value of exploring the drug combinations in normal immortalized cell lines such as MCF10 and normal ovarian cell lines, we do not have the resources to test this hypothesis currently.
- As BRCA1 is downregulated, have the authors checked for the involvement of low fidelity polymerases? Given the downregulation of BRCA1, the authors should investigate the expression and activity of low fidelity polymerases, which might be upregulated to compensate for the impaired DNA repair and could contribute to genomic instability and resistance.
Reply: The reviewer raises an important point regarding the potential involvement of low fidelity polymerases in response to BRCA1 downregulation. We added the following sentences in the limitations section of the Discussion: “With downregulation of BRCA1, it is conceivable that the expression and activity of low fidelity polymerases might be upregulated to compensate for the impaired DNA repair and could contribute to genomic instability and resistance [38].” (Page 10; line no. 284 - 286).
Unfortunately, we do not have additional resources to test this hypothesis currently.
- Based on the evidence of c-MYC downregulation and DNA repair impairment, what is the role of AXL and low fidelity polymerases? The authors should discuss the potential roles of AXL and low fidelity polymerases in the context of their findings. AXL is known to be involved in cancer progression and resistance and generation of persister cells, while low fidelity polymerases may further compromise DNA repair mechanisms.
Reply: To address the reviewer’s comment, we added the following paragraph in the Discussion section:
“While our study provides evidence of c-MYC downregulation and DNA repair impairment, the specific roles of AXL and low-fidelity polymerases in this context warrant further investigation. The critical role of AXL, a receptor tyrosine kinase, in promoting tumor cell survival, metastasis, and therapeutic resistance is well established [43,44]. AXL activation can lead to the generation of persister cells, a subpopulation of cancer cells that can survive treatments and potentially cause tumor recurrence [43,45-47]. Therefore, AXL may influence the response to combination therapy involving HDACis, PARPis, and decitabine. AXL signaling pathways could potentially interact with the mechanisms targeted by these drugs, affecting their efficacy in inhibiting cell proliferation, inducing apoptosis, and impairing DNA repair pathways. Similarly, the presence of low-fidelity polymerases could further disrupt DNA repair mechanisms targeted by PARPis, potentially enhancing the cytotoxic effects of the combination treatment. Furthermore, low-fidelity polymerases such as Pol η, Pol ι, Pol κ, and POLQ, which are involved in translesion DNA synthesis (TLS), may introduce mutations ow-ing to their error-prone nature, further compromising DNA repair mechanisms, and contributing to genomic instability [48-50]. In the presence of c-MYC downregulation, the reliance on low-fidelity polymerases may increase, leading to a higher tumor mu-tational burden, cancer progression and resistance to treatment. Further research is needed to elucidate the role of AXL signaling, low-fidelity polymerase activity, and impaired DNA repair for developing targeted therapies in the clinical setting.” (Page 11; line no. 329 - 347).
- Authors have to validate the synergy effects in animal models? To enhance the scientific soundness of the study, the authors should conduct in vivo experiments using at least one xenograft model from each cancer cell line. Testing the drug combination that showed the strongest synergy in these models would provide valuable insights into the therapeutic potential of the combination treatments.
Reply: We agree with the reviewer that the synergy effects presented here should be validated in animal models. Unfortunately, we do not have the resources to perform these experiments. We added the following sentences in the limitation part of the Discussion section. “In vivo experiments with xenograft models from each cancer cell line were not per-formed. Although animal models are frequently used, they have their own limitations. They are costly and lengthy, and results derived from these studies cannot be directly extrapolated to humans, owing to differences in the metabolism of the study drugs between humans and animal models. Carefully designed phase I-II studies are still required to confirm if the obtained results are clinically meaningful.” (Page 11; Lines 318 - 324).
- Authors have to analyze the impact of the drug synergy on the cell cycle? Given the strong synergistic effects observed, the authors should perform a cell cycle analysis to determine how the drug combinations affect cell cycle progression and contribute to the cytotoxic effects.
Reply: To address the reviewer's comment, we added the following in the limitations part of the Discussion section: “Cell cycle analyses to determine how the drug combinations affect cell cycle progression and contribute to the cytotoxic effects [42] were not performed.” (Page 11; line no. 307 – 309).
Unfortunately, we do not have the resources to test this hypothesis currently.
- Authors have to include p-values in all the graphs (Including in the figure will be helpful for the readers to understand the comparisons. Also include the number of replicates in the figure legend.
Reply: We agree with the reviewer that the p-values in the Figure are important. We discussed this request with the statistician and co-author of the paper, Dr Clark R. Andersen, who stated that “in addition to the confidence intervals, the significance of the Figures is indicated by solid-filled circles, as noted in the captions, and all p-values are provided in Table 1 and Supplemental Tables. Including the p-values within the Figures would add complexity to the already complex figures”.
We have added the number of replicates in the figure legend as recommended.
Minor Comments:
- Ensure the correct spelling for c-MYC (not c-MYK) in figure 4B.
Reply: Thank you for pointing out the mistake. We have corrected the spelling.
Round 2
Reviewer 2 Report
Comments and Suggestions for Authors
The authors have addressed the comments textually; however, they have missed important references. AXL and TLS have not been discussed in any of the papers cited (45-50). Additionally, several crucial references relevant to the field are missing. The authors must include all relevant references.
Author Response
Comment: The authors have addressed the comments textually; however, they have missed important references. AXL and TLS have not been discussed in any of the papers cited (45-50). Additionally, several crucial references relevant to the field are missing. The authors must include all relevant references.
Response: Thank you for your thorough review and constructive feedback on our manuscript. To address the reviewer’s comments, we have included the following relevant references. If the reviewer has recommendations for additional references, we would appreciate it if s/he could indicate those specific references that we should include in our manuscript.
- Balaji, K.; Vijayaraghavan, S.; Diao, L.; Tong, P.; Fan, Y.; Carey, J.P.; Bui, T.N.; Warner, S.; Heymach, J.V.; Hunt, K.K.; et al. AXL Inhibition Suppresses the DNA Damage Response and Sensitizes Cells to PARP Inhibition in Multiple Cancers. Mol Cancer Res 2017, 15, 45-58, doi:10.1158/1541-7786.MCR-16-0157.
- Ramirez, M.; Rajaram, S.; Steininger, R.J.; Osipchuk, D.; Roth, M.A.; Morinishi, L.S.; Evans, L.; Ji, W.; Hsu, C.H.; Thurley, K.; et al. Diverse drug-resistance mechanisms can emerge from drug-tolerant cancer persister cells. Nat Commun 2016, 7, 10690, doi:10.1038/ncomms10690.
- Mikubo, M.; Inoue, Y.; Liu, G.; Tsao, M.S. Mechanism of Drug Tolerant Persister Cancer Cells: The Landscape and Clinical Implication for Therapy. J Thorac Oncol 2021, 16, 1798-1809, doi:10.1016/j.jtho.2021.07.017.
- Yang, W.; Woodgate, R. What a difference a decade makes: insights into translesion DNA synthesis. Proc Natl Acad Sci U S A 2007, 104, 15591-15598, doi:10.1073/pnas.0704219104.
- Vaziri, C.; Rogozin, I.B.; Gu, Q.; Wu, D.; Day, T.A. Unravelling roles of error-prone DNA polymerases in shaping cancer genomes. Oncogene 2021, 40, 6549-6565, doi:10.1038/s41388-021-02032-9.
- Ziv, O.; Zeisel, A.; Mirlas-Neisberg, N.; Swain, U.; Nevo, R.; Ben-Chetrit, N.; Martelli, M.P.; Rossi, R.; Schiesser, S.; Canman, C.E.; et al. Identification of novel DNA-damage tolerance genes reveals regulation of translesion DNA synthesis by nucleophosmin. Nat Commun 2014, 5, 5437, doi:10.1038/ncomms6437.